# *Talaromyces purpurogenus* Isolated from Rhizosphere Soil of Maize Has Efficient Organic Phosphate-Mineralizing and Plant Growth-Promoting Abilities

Xuefang Sun [1,2,†], Feng Liu [1,†], Wen Jiang [1,2], Peiyu Zhang [1], Zixuan Zhao [1], Xiang Liu [1], Yan Shi [1] and Qing Sun [1,2,*]

1   Shandong Provincial Key Laboratory of Dryland Farming Technology, College of Agronomy, Qingdao Agricultural University, Qingdao 266109, China
2   Academy of Dongying Efficient Agricultural Technology and Industry on Saline and Alkaline Land in Collaboration with Qingdao Agricultural University, Dongying 257091, China
*   Correspondence: sunqing@qau.edu.cn
†   These authors contributed equally to this work.

**Abstract:** The scarcity of phosphorus (P) makes improving phosphorus use efficiency a critical issue in crop production. Plant rhizosphere microorganisms play a vital role in increasing phosphorus bioavailability and promoting the level of plant-absorbable P in agroecosystems. In this study, *Talaromyces purpurogenus* SW-10 strain with efficient organic phosphate-mineralizing ability was isolated from maize rhizosphere soil. SW-10 showed efficient phytate utilization with corresponding soluble P levels of 525.43 mg/L and produced phytase in the liquid medium. The response surface methodology (RSM) analysis showed that glucose as the carbon source and $(NH_4)_2SO_4$ as the nitrogen source at 28 °C and pH 7.0 promoted higher mineralization of insoluble organic phosphate. When cocultivated with different genotypes of maize seedlings, SW-10 significantly increased the shoot's dry weight by 37.93%, root's dry weight by 31.25%, and the plant height by 13.03% for low-P sensitive inbred line 31778, while no significant change was observed in the low-P tolerance inbred line CCM454. In addition, SW-10 strain significantly increased total P-concentration in the shoots (22.4%~32.9%) and roots (3.10%~9.77%) for both inbred lines. In conclusion, the isolated *T. purpurogenus* SW-10 strain possesses an efficient organic phosphate-mineralizing ability and maize plant growth-promoting effect, especially for the low-P sensitive genotype that could be exploited for enhancing P availability in agriculture.

**Keywords:** organic phosphate-mineralizing fungi; phytase; maize rhizosphere; plant growth-promotion

## 1. Introduction

Phosphorus (P) is an indispensable component of enzymatic reactions, signal transduction processes, and energy metabolisms, and thus is a primary essential macronutrient for the normal growth and development of plants [1]. Although P is abundant in soil, only a minuscule amount of soluble P is directly assimilated by plants since a large portion of P exists in an inorganic or organic insoluble form [2,3]. To obtain high yields, an excessive quantity of P fertilizer is added to the farmer's land. Excessive application of external P has led to low utilization efficiency, numerous environmental issues such as water pollution and soil compaction, and an increasing economic burden [4,5]. An effective strategy for overcoming these issues is to transform insoluble P into plant-absorbable orthophosphate via P-solubilizing microbes (PSMs).

PSMs are widely distributed in soil, freshwater, seawater, and sediments, and are responsible for cycling insoluble P to soluble $PO_4^{3-}$ ions [6,7]. Compared with bacteria, phosphate-solubilizing fungi (PSF) possess a higher ability to release P from insoluble phosphate compounds [8]. In addition, it is more stable for fungi to maintain their genetic

traits than for bacteria. Soil PSFs, particularly filamentous fungi, have been extensively studied, such as *Aspergillus* and *Penicillium* [9]. Doilom et al. isolated 1 PSF from air and later identified an airborne strain (KUMCC 18-0196) as *Aspergillus hydei*, which was found to be capable of solubilizing phosphates in liquid PVK medium [10]. *Penicillium oxalicum* PSF-4 showed exceptional solubilization of tricalcium phosphate (TP) and iron phosphate (IP) [11]. PSF also promotes plant growth; for instance, soybean plants showed significant growth, root nodulation, and yield increase after being inoculated with PSF *Aspergillus niger* [12]. Nevertheless, the majority of the research studies on PSFs are on the insoluble inorganic P solution instead of the insoluble organic P mineralization.

Organic P, which can account for up to 80% of soil P, occurs principally in the form of phytate [13]. This is particularly true for soils that are cultivated in high tunnels, which make soil quality and fertility sustainable. Manure-based compost used heavily in high tunnel farming can lead to excessive accumulations of insoluble organic P in soil. As phytate cannot be directly used by plants, it is strongly absorbed by the soil, reducing its bioavailability [14]. It has been suggested that microbial phytases may play an important role in mineralizing P from phytate in soil and making it available to plants for uptake [15]. Therefore, exploring more organic phosphate mineralizing microbes to enhance soil P nutrition is urgent.

Maize is a major staple cereal and the highest-produced crop. In addition to being a global food, it is also an important raw material for energy production, as well as having a variety of industrial uses [16]. Numerous plant growth-promoting microorganisms are present in the rhizosphere soil of maize, which promotes plant growth and crop production [17]. In addition, increasing evidence has indicated that fungal responsiveness is highly variable among plant genotypes [18]. Thus, PSF isolation in the rhizosphere determines its overall P-solubilization capacity and its ability to promote plant growth.

In this study, we focused on the isolation of organic phosphate mineralizing microbes from the rhizosphere soil of maize. The isolates showing high P mineralizing abilities were optimized to maximize the phosphate-mineralizing activity using the response surface methodology (RSM) method, which is a valuable tool to optimize the culture parameters. In addition, the effects of the isolates on the different genotypes of low-P sensitive and low-P tolerance maize growth under pot culture conditions were investigated. The objective of this study was to provide new potential strains for biofertilizer research and increase agricultural productivity.

## 2. Materials and Methods

### 2.1. Soil Sampling

Soil samples were collected from a maize field located in Qingdao City (36.09° N, 119.90° E), Shandong Province, China. Maize plants at the mature stage were carefully uprooted using a shovel; afterwards, loosely attached soil was shaken off vigorously. The rhizosphere soil adhering to plant roots was collected using a sterile brush and transferred to sterilized bags, which were placed in an icebox and immediately transported to the laboratory. The rhizosphere soil showed the following characteristics: 66.17 mg kg$^{-1}$ available phosphorus (AP), 0.94 g kg$^{-1}$ total nitrogen (TN), 162.85 mg kg$^{-1}$ available potassium (AK), and 16.58 g kg$^{-1}$ organic matter (SOM), pH 5.6 (soil: water in 1:1 ratio).

### 2.2. Isolation and Identification of Organic Phosphate-Mineralizing Fungi

Soil samples (10 g) were added individually to 250-mL flasks with 90 mL sterile distilled water. The mixture was shaken at 200 r/min for 30 min at 28 °C. Then, 100 μL of serially diluted ($10^{-4}$–$10^{-6}$) solution was inoculated into the modified NBRIP agar-solidified medium (10 g/L glucose, 0.5 g/L $(NH_4)_2SO_4$, 0.3 g/L NaCl, 0.3 g/L KCl, 0.3 g/L $MgSO_4·7H_2O$, 0.03 g/L $FeSO_4·7H_2O$, 0.03 g/L $MnSO_4·H_2O$, 2.5 g/L calcium phytate, 15 g/L agar), and incubated at 28 °C for 4–6 days. Colonies with clear halos were selected.

### 2.3. DNA Extraction, PCR Amplification, and Phylogenetic Analysis

The fungal isolates were identified based on routine cultural and morphological characteristics and microscopical features in the solid PDA medium. For molecular identification, the liquid PDA medium was used to grow the mycelia for two days before harvesting. These mycelia were extracted using CTAB to extract genomic DNA. The rDNA internal transcribed spacer (ITS) and β-tubulin fragments were amplified using universal primers of ITS1 (5-TCCGTAGGTGAACCTGCGG-3)/ITS4 (5-TCCTCCGCTTATTGATATGC-3) and Bt2a (5-GGTAACCAAATCGGTGCTGCTTTC-3)/Bt2b (5-ACCCTCAGTGTAGTGACCCTTGGC-3). The PCR mixture contained: 2× PCR mixture (12.5 μL) (TAKARA), 10 μM primers (1 μL), template DNA (2 μL), and ddH$_2$O (8.5 μL). The PCR was performed at 94 °C for 5 min, followed by 35 cycles of 94 °C for 30 s, 55 °C for 30 s, and 72 °C for 1 min, and final extension at 72 °C for 10 min. PCR products were sized on 1% agarose gel and sequenced on Sangon Biotech. A phylogenetic tree was constructed with MEGA version 5.1 software using the neighbor-joining method.

### 2.4. Organic Phosphate-Mineralization Efficiency in Liquid Medium

The organic phosphate-mineralization ability of the fungal isolates was evaluated using the liquid NBRIP growth medium. Except for agar, the composition of the liquid NBRIP medium formula is identical to that of solid medium. Inoculated medium was incubated at 28 °C for five days at 200 rpm in the dark. Fungal culture was centrifuged at 6000 rpm for 10 min to obtain the supernatant. A molybdenum blue method was used to determine the concentration of soluble phosphorus in the supernatant [10]. Briefly, the process is as follows: 1.0 mL of the supernatant was pipetted into a 50 mL calibrated flask, water was added to 35 mL, and the solution was mixed well. Two drops of dinitrophenol indicator and 4 mol L$^{-1}$ NaOH solution were added until the solution turned yellow. Then, 5 mL molybdenum-antimony reagent was added, and water was added to 50 mL. The mixture was then color-rendered at 25–30 °C for 30 min. Finally, the light absorbance of this mixture was measured using a UV-Vis spectrophotometer at 700 nm. Phytase activity was determined in 0.2 mL of culture filtrate incubated with 1.8 mL of acetic acid buffer at 37 °C for 5 min, added to 4 mL substrate solution (prepared in 0.25 M sodium acetate buffer of pH 5.5 containing 0.75 mM sodium phytate), mixed thoroughly, hydrolyzed at 37 °C in a water bath for 30 min, and finally, color-developing solution was added to this mixture. A UV-Vis spectrophotometer Cary60 was used to measure the light absorbance of this mixture at 415 nm [19].

### 2.5. Optimization of Fermentation Parameters by Single-Factor Experiments

Carbon sources, such as glucose, D-fructose, α-maltose, sucrose, α-lactose, and soluble starch, were used to evaluate their effects on the phosphate-mineralizing ability. The original carbon source in NBRIP was replaced with these sources at 1% (w/v) concentrations. In addition, various nitrogen supplements, such as ammonium sulfate, urea, potassium nitrate, and ammonium chloride were tested to examine their effects on phosphate-mineralizing ability. Moreover, culture conditions were optimized for different parameters, including the initial pH (4.0, 5.0, 6.0, 7.0, and 8.0) and fermentation temperature (24 °C, 26 °C, 28 °C, 30 °C, and 32 °C), to explore the organic phosphate-mineralizing ability.

### 2.6. Screening of Significant Fermentation Parameters by the RSM

The experimental design consisted of two sequential steps. Firstly, the categorical factors were analyzed to determine the best combination of carbon and nitrogen sources. Then, a Central Composite Design (CCD) combined with RSM was applied to optimize the organic phosphate-mineralizing conditions. Based on the outcomes of the single-factor experiment, the effects of temperature and initial pH were screened to enhance the organic phosphate-mineralizing ability in the preliminary study. Factors are coded using their coded values according to the CCD. The test levels of the independent variables were coded with −1.414, −1, 0, 1, 1.414 (Table 1), and a total of eight factorial points, and five

central points were designed. RSM was performed to analyze the experimental design using the software (Design Expert software, version 10.7, Stat-Ease Inc., Minneapolis, MN, USA). Fitting Equation (1) was performed using a six-coefficient quadratic model, where Y represents the production of available P content, while A and B are independently evaluated factors (coded variables), $b_0$ is the intercept, and $b_i$ is the parametric coefficient in multiple regression:

$$Y = b_0 + b_1A + b_2B + b_3AB + b_4A^2 + b_5B^2 \tag{1}$$

**Table 1.** Parameter and level of the Central Composite Design.

| Parameter | Level | | | | |
|---|---|---|---|---|---|
| | −1.414 | −1 | 0 | 1 | 1.414 |
| A | 4.586 | 5 | 6 | 7 | 7.414 |
| B | 25.172 | 26 | 28 | 30 | 30.828 |

A: pH; B: Temperature (°C).

### 2.7. Pot-Experiment Design

A pot experiment was performed with the low-P sensitive (31778) and low-P tolerance (CCM454) genotypes of maize with five replicates for each treatment to determine whether the isolates have any effect on plant growth. The criteria for distinguishing between low-P sensitive and low-P tolerance genotypes have been described in the previous article published by our laboratory [20]. Maize seeds were surface-sterilized with 3% NaClO for 20 min and then washed three times with distilled water. The seeds were germinated on sterile Petri dishes containing moist filters in a constant temperature incubator at 28 °C for 2–3 days. Each pot (16.5 cm diameter, 14.5 cm high) was supplemented with 200 g of air-dried vermiculite. Calcium phytate as an insoluble phosphate source was blended with the vermiculite.

Homogenous seedlings were sown in plastic pots and grown in the greenhouse. For the SW-10 treatment, 1 mL of a $10^7$ CFU/mL spore suspension of the isolated fungi was added to the vermiculite, while an equal amount of sterile water was added as the control to each pot. All the pots were irrigated with nutrient solution in the absence of $KH_2PO_4$. Maize seedlings were grown for three weeks, and whole plants were harvested and used to determine plant biomass, total P concentration, and soil's available P content. The various parameters were determined using standard analytical methods as reported previously [14].

### 2.8. Statistical Analysis

Data analysis was carried out using SAS 9.4 using ANOVA followed by Duncan's test ($p < 0.05$). Charts were made using Excel (Microsoft Office Excel 2019).

## 3. Results

### 3.1. Isolation and Identification of Organic Phosphate-Mineralizing Fungi

A clear zone around microbial colonies on the NBRIP medium indicates the organic phosphate-mineralizing ability of the microorganisms. In this study, eight fungal strains were isolated from the maize rhizosphere soil demonstrating organic phosphate-mineralizing characteristics. Among the isolates, SW-10 exhibited the highest organic phosphate-mineralizing ability on insoluble organic P medium (Figure 1A). Therefore, the SW-10 strain was selected for further experiments to determine its ability to mineralize insoluble organic P.

The primary hyphae of SW-10 were white in color, which gradually turned orange on the PDA medium. After seven days of incubation in the dark at 28 °C, dark green conidia appeared in the center of the colony, orange hyphae in the margin, and red pigments at the bottom (Figure 1B). Optical microscopes were used to examine the spores and conidiophores of the PSF. SW-10 presented typical penicillate conidiophores with conidia.

Based on the phylogenetic analysis of rDNA ITS and β-tubulin region sequence, the SW-10 strain was identified as *Talaromyces purpurogenus* (Figure 1C), and was deposited at the China General Microbiological Culture Collection Center (CGMCC) (accession number: CGMCC No. 20737).

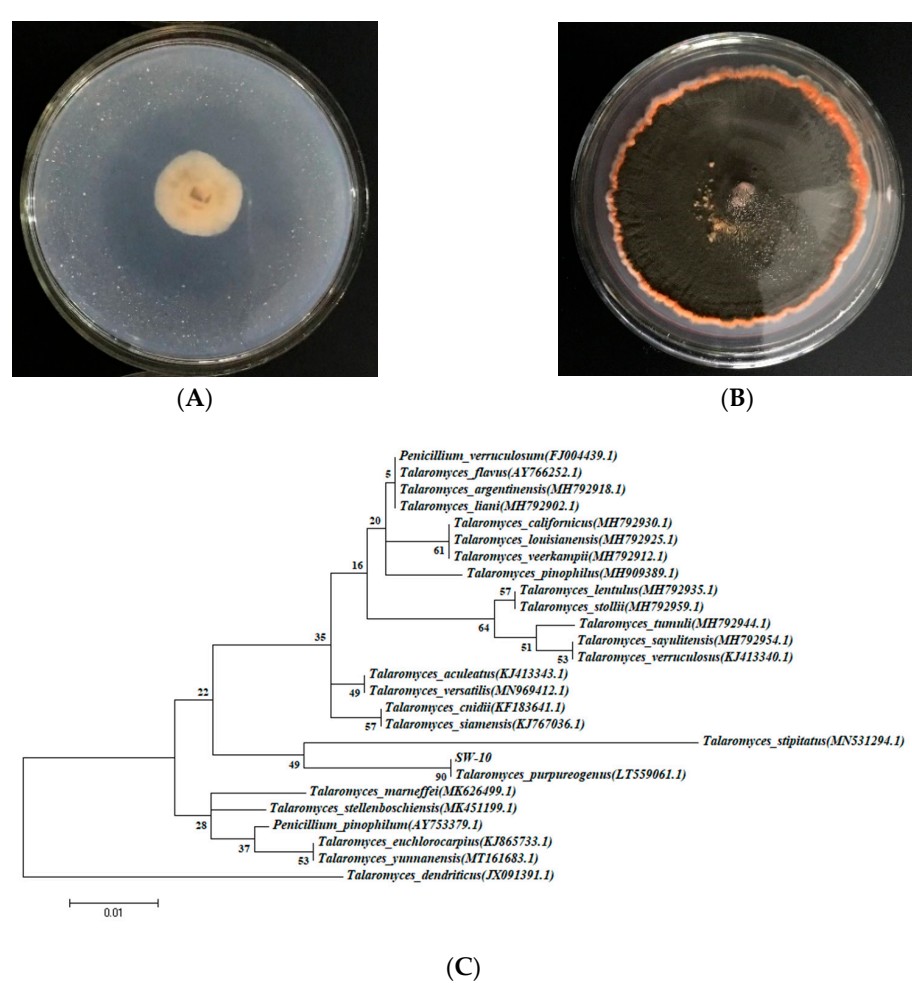

**Figure 1.** Zones of phosphate solubilization on NBRIP agar plates produced by SW-10 (**A**), colonies of SW-10 on PDA at 28 °C after 7-day incubation (**B**), and a phylogenetic tree of the β-tubulin sequence of SW-10 (**C**).

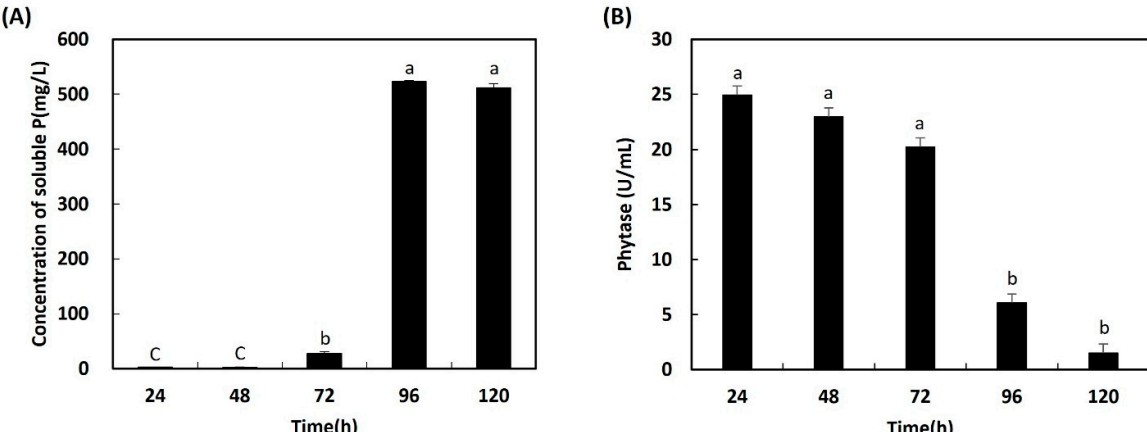

**Figure 2.** Concentrations of soluble P mineralized by SW-10 strain (**A**) and changes in phytase activity (**B**). The different letters above the SD bars indicate very significant difference at $p \leq 0.05$.

### 3.2. Organic Phosphate-Mineralization Ability of the SW-10 Strain

In this study, SW-10 strain was inoculated into the NBRIP liquid medium to determine its ability to mineralize organic phosphate. In the early growth period, the soluble P content increased slowly; after 72 h, the content of the soluble P rapidly increased to 523.12 mg/L. Then, the maximum soluble P content remained unchanged (Figure 2A). The phytase activity was found to be approximately 23 U/mL in 1–3 days of culture, which decreased after 4–5 days (Figure 2B).

### 3.3. Optimization of Organic Phosphate-Mineralizing Conditions by Single-Factor Experiments

To investigate the optimum phosphate-solubilizing conditions of SW-10, four key parameters ((i) carbon source, (ii) nitrogen source, (iii) pH, and (iv) incubation temperature,) were individually studied using a single-variable approach to evaluate the effect of each condition on the organic phosphate-mineralizing ability. The organic phosphate-mineralizing ability of SW-10 under different carbon source culture conditions was significantly different (Figure 3A). In different carbon source media, the ability to mineralize phosphate was found to be in the following descending order: glucose > starch > maltose > D-fructose > sucrose > α-lactose. When glucose and α-lactose were used as the sole carbon sources, the effective P contents reached 524.17 mg/L and 2.67 mg/L, respectively. Different nitrogen sources and ammonium sulfate, urea, and potassium nitrate treatments had no significant effect on the organic P mineralization ability of SW-10 (Figure 3B).

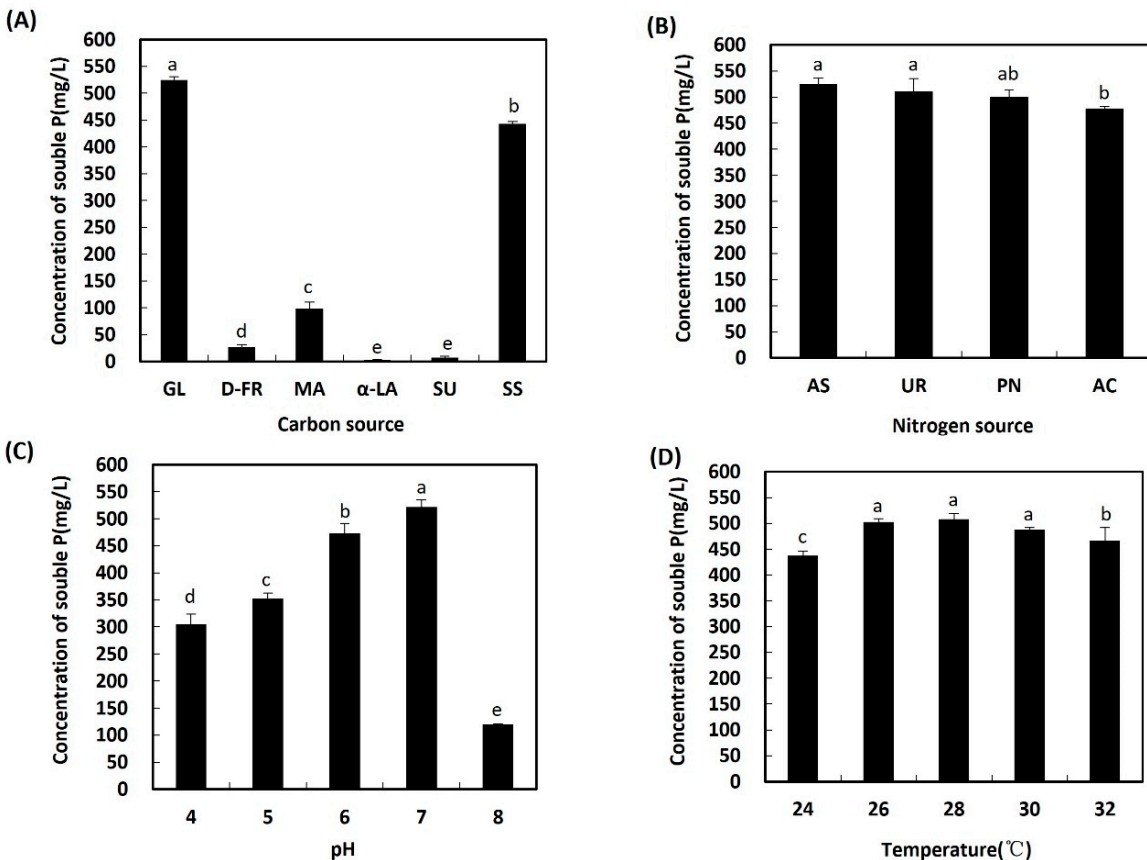

**Figure 3.** Effects of different culture conditions including carbon source (**A**), nitrogen source (**B**), pH (**C**) and temperature (**D**) on the phosphate-mineralizing capacity of SW-10 strain. Carbon source: GL: glucose, D-FR: D-fructose, MA: maltose, α-LA: α-lactose, SU: sucrose, SS: soluble starch; nitrogen source: AS: ammonium sulfate, UR: urea, PN: potassium nitrate, AC: ammonium chloride. The different letters above the SD bars indicate very significant difference at $p \leq 0.05$.

However, differences in pH had a significant impact on the organic phosphate mineralization ability of SW-10 (Figure 3C). At pH 7.0, SW-10 strain's phosphate mineralization ability reached 521.84 mg/L, whereas at pH 8.0, P concentration was 119.83 mg/L. The temperature had no significant effect on the P mineralization ability of SW-10 (Figure 3D). The P mineralization ability of SW-10 did not vary significantly in the temperature range of 26–30 °C. At 32 °C, the P mineralization ability of the strain decreased marginally, indicating the wide temperature adaptability of the SW-10 strain.

### 3.4. Optimizing Organic Phosphate-Mineralizing Parameters by RSM

A categorical experimental design was applied to select the best combination of carbon and nitrogen sources to optimize organic phosphate-mineralizing ability. As per the outcomes of this experiment, glucose and $(NH_4)_2SO_4$-based media showed the highest P mineralization, followed by urea and glucose, whereas the P mineralization was very low in $KNO_3$ and $\alpha$-lactose (Table 2). Thus, the results indicated that the best combination of glucose and $(NH_4)_2SO_4$ was used in the experiments. The scheme of the response surface methodology and experimental results is depicted in Table 3. The regression equation of the P-mineralization amount of SW-10 strain was obtained by using multiple regressions of fitting score analysis (2), where A is the initial pH, B is the temperature, and Y is the production of available P content. The present model and data analysis allowed us to define the optimal media composition for P mineralization.

$$Y = 471.7 + 69.74A + 8.72B - 1.74AB - 36.82A^2 - 16.2B^2 \tag{2}$$

**Table 2.** The best combination of carbon source (mg $L^{-1}$) and nitrogen source (mg $L^{-1}$), influencing P-mineralization ability.

| Carbon Source | Nitrogen Source | | | |
|---|---|---|---|---|
| | $(NH_4)_2SO_4$ | Urea | $KNO_3$ | $NH_4Cl$ |
| Glucose | 525.43 | 502.65 | 494.91 | 474.29 |
| D-Fructose | 31.61 | 26.72 | 7.85 | 27.35 |
| Maltose | 70.91 | 62.43 | 27.51 | 65.26 |
| Sucrose | 11.50 | 7.42 | 2.93 | 7.31 |
| $\alpha$-Lactose | 2.48 | 1.65 | 1.20 | 2.20 |
| Starch | 450.52 | 456.53 | 411.04 | 432.68 |

**Table 3.** Scheme of response surface methodology and experimental results.

| No. | Run | A | B | Y |
|---|---|---|---|---|
| 1 | 6 | −1 | −1 | 341.67 |
| 2 | 2 | 1 | −1 | 477.51 |
| 3 | 9 | −1 | 1 | 350.06 |
| 4 | 1 | 1 | 1 | 492.86 |
| 5 | 12 | −1.414 | 0 | 302.49 |
| 6 | 11 | 1.414 | 0 | 499.98 |
| 7 | 4 | 0 | −1.414 | 426.2 |
| 8 | 10 | 0 | 1.414 | 458.75 |
| 9 | 3 | 0 | 0 | 473.47 |
| 10 | 13 | 0 | 0 | 482.16 |
| 11 | 8 | 0 | 0 | 468.17 |
| 12 | 5 | 0 | 0 | 471.56 |
| 13 | 7 | 0 | 0 | 463.16 |

A indicates pH; B indicates temperature; Y indicates phosphate solubilization capacity (mg·$L^{-1}$).

The outcome of the ANOVA for the quadratic model was summarized in Table 4. The values of $R^2$ and Adj $R^2$ were found to be 0.993 and 0.988, respectively. The desirable determination coefficients indicated a high correlation between the experimental results and

predicted response values. Meanwhile, the value of lack of fit ($p > 0.05$) was not significant, and the level of the model was highly significant ($p < 0.01$). The results suggested that the variation could accurately predict the mode.

**Table 4.** Analysis of variance with glucose as the sole carbon source.

| Source | Sum of Squares | df | Mean Square | F-Value | *p*-Value |
|---|---|---|---|---|---|
| Model | 49,880.18 | 5 | 9976.04 | 205.23 | <0.0001 |
| A | 38,911.16 | 1 | 38,911.16 | 800.48 | <0.0001 |
| B | 608.53 | 1 | 608.53 | 12.52 | 0.0095 |
| AB | 12.11 | 1 | 12.11 | 0.25 | 0.633 |
| A2 | 9429.51 | 1 | 9429.51 | 193.98 | <0.0001 |
| B2 | 1824.99 | 1 | 1824.99 | 37.54 | 0.0005 |
| Residual | 340.27 | 7 | 48.61 | | |
| Lack of Fit | 142.31 | 3 | 47.44 | 0.96 | 0.4935 |
| Pure Error | 197.96 | 4 | 49.49 | | |
| Cor Total | 50,220.45 | 12 | | | |
| | | | | R-Squared | 0.993 |
| | | | | Adj R-Squared | 0.988 |
| | | | | C.V.% | 1.59 |

Figure 4 represented the contour plot and the 3D-response surface graphs depicting the regression model for available phosphorous content by the fungus *T. purpurogenus* SW-10 strain, demonstrating the interaction between two variables to determine the optimum level of each variable for maximum response. The contour plot demonstrated that both temperature and pH had the maximum value and were within the test range. The 3D response surface showed that as the temperature changes, the response value first increased and then decreased. Similarly, with increasing pH, the response value first increased and then decreased. The optimum values of the two factors selected for the fermentation process were obtained by solving Equation (1) using the Design-Expert software package. The optimal phosphorate solubilizing conditions of the *T. purpurogenus* SW-10 strain were statistically predicted as follows: glucose as the carbon source, $(NH_4)_2SO_4$ as the nitrogen source, at 28 °C and pH 7.0. Under this condition, the available P content was predicted to be 506.39 mg/L. To verify the optimal P solubilization ability conditions obtained by the response surface method, the experimental result was 520.08 mg/L, which was consistent with the previous results.

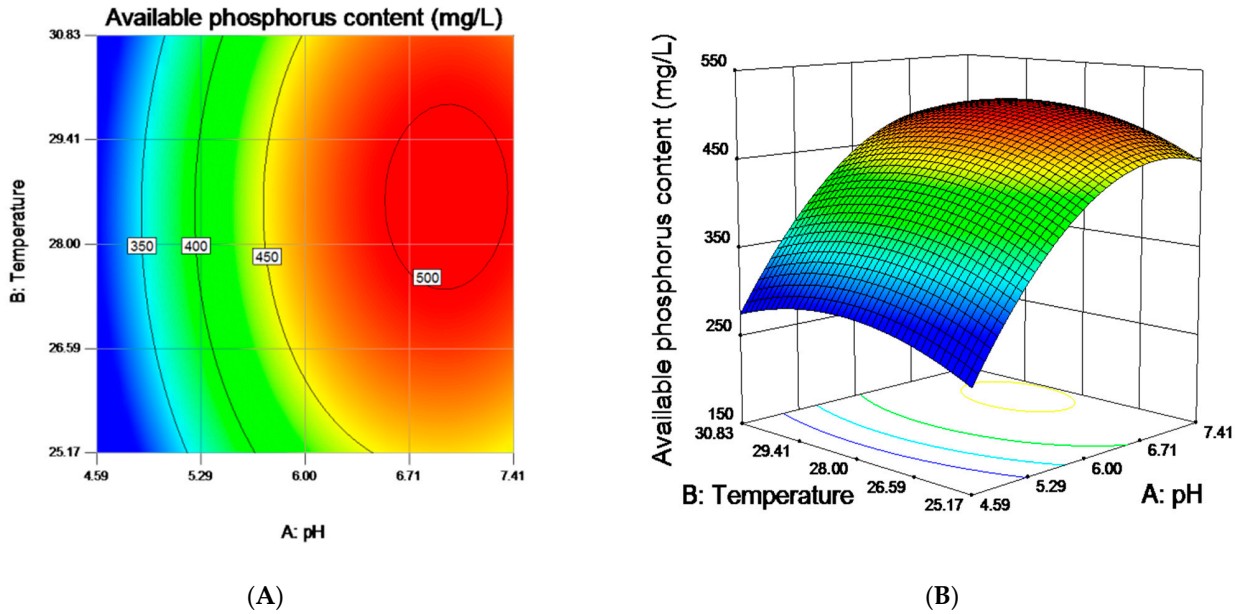

**(A)**　　　　　　　　　　　　　　　　　　　　　　　　**(B)**

**Figure 4.** Contour plot (**A**) and response surface plot (**B**) for available phosphorus content by SW-10.

### 3.5. Effects of T. purpurogenus on Different Genotypes of Maize Seedling Growth

As per the outcomes of the above-mentioned experiments, the SW-10 strain showed a strong ability to mineralize insoluble organic phosphate. Thus, the effect of SW-10 strain on plant growth was also explored. SW-10 strain was applied on the roots of different genotypes of maize seedlings, and the growth status of these seedlings was monitored. Growth responsiveness differed between low-P tolerant inbred line CCM454 and low-P sensitive inbred line 31778 when the plants were colonized with SW-10 strain (Table 5). For low-P sensitive inbred line 31778, SW-10 significantly affected the growth parameters where shoot fresh and dry weight increased by 55.95% and 37.93%, root fresh weight and dry weight increased by 31.20% and 31.25%, and plant height increased by 13.03% compared to the control group. These differences were not significant for low-P tolerant inbred line CCM454. These results indicated different responsiveness results of maize genotypes to SW-10 strain under low P conditions.

**Table 5.** Effect of inoculation with SW-10 on the growth of maize seedlings.

| Maize Inbred Line | Treatment | Shoot | | Root | | Plant Height (cm) |
| | | Fresh Weight (g/Plant) | Dry Weight (g/Plant) | Fresh Weight (g/Plant) | Dry Weight (g/Plant) | |
|---|---|---|---|---|---|---|
| 31778 | Control | 3.11 b | 0.29 b | 2.50 b | 0.16 b | 23.03 b |
| | SW-10 | 4.85 a | 0.40 a | 3.28 a | 0.21 a | 26.03 a |
| CCM454 | Control | 2.41 a | 0.23 a | 2.10 a | 0.15 a | 24.02 a |
| | SW-10 | 2.73 a | 0.26 a | 2.45 a | 0.15 a | 24.62 a |

Means within a column followed by various letters are significantly different ($p < 0.05$)

### 3.6. Effects of T. purpurogenus on Total P Concentration of Plants and Soil's Available P Content

SW-10 strain led to a significant increase in P concentration in the 31778 and CCM454 inbred lines (Table 6). For the shoot, the P concentration increased by 32.9% for 31778 and by 22.4% for CCM454 compared with control. For the root, the P-concentration increased by 9.77% for 31778 and by 3.10% for CCM454 compared with control. However, there was no significant difference in soil's available P content between the soil inoculated with strain SW-10 and the noninoculated control. These results indicated that SW-10 increased the plant total phosphorus content and had a larger effect on low-P sensitive inbred line 31778.

**Table 6.** Effects of SW-10 on plant's total P concentrations and soil's available phosphorus content.

| Maize Inbred Line | Treatment | Total Phosphorus Content (g/kg) | | Soil Available Phosphorus Content (mg/kg) |
| | | Shoot | Root | |
|---|---|---|---|---|
| 31778 | Control | 3.53 b | 2.66 b | 62.72 a |
| | SW-10 | 4.69 a | 2.92 a | 61.36 a |
| CCM454 | Control | 3.40 b | 2.58 b | 59.65 a |
| | SW-10 | 4.16 a | 2.66 a | 60.91 a |

Means within a column followed by various letters are significantly different ($p < 0.05$).

## 4. Discussion

### 4.1. T. purpurogenus SW-10 with Organic Phosphate-Mineralizing Ability Was Successfully Isolated from Maize Rhizosphere Soil

P is an essential macroelement for anabolisms and energy-producing metabolisms in plants. It is also one of the main fertilizers for high-yield agricultural production [21]. Due to the limitations of microbial ecological adaptability, the utilization of native microorganisms in developing biological fertilizer has shown obvious advantages [22]. Several phosphate-solubilizing microorganisms, such as actinomyces, bacteria, and fungi, have been screened for phosphate solubilization. They were shown to enhance the solubilization of insoluble P compounds [23,24]. While the number of PSB far exceeds that of PSF, the phosphate-

solubilizing capability of fungi is generally superior to that of bacteria and can be several dozen times higher [25]. Therefore, it is important to isolate PSM that can efficiently dissolve phosphate and promote plant growth in rhizosphere soil.

In this study, six PSF strains were isolated from the rhizosphere soil of maize, of which *T. purpurogenus* SW-10 strain showed the highest mineralizing ability for insoluble organic phosphate. *T. purpurogenus* was previously described as a plant endophyte, indicating its close relationship with the plant root [26]. Previous studies have shown that these microorganisms produce terpenoids, red pigments, antiproliferative, and antioxidative bioactive compounds [27,28], but they have not been previously described as solubilizers of 'unavailable' forms of P. In this study, for the first time, *T. purpurogenus* was reported as organic phosphate-solubilizing fungi. Previous research studies on PSF were primarily focused on the solution of insoluble inorganic P. The use of microorganisms for organic P recycling has not been widely explored [29]. As organic fertilizers replace chemical fertilizers, there is more insoluble organic P in the soil. There is a need to explore more insoluble organic phosphate-mineralizing fungi.

### 4.2. Production of Phytase Activities Is the Core Mechanism of SW-10 during Organic Phosphate Mineralization

Organic phosphate is primarily stored as phytate; however, the plants cannot utilize it, as phytate forms a complex with cations [30]. In this study, calcium phytate was used as a source of insoluble organic phosphate. At the early stages of culture, the supernatant contained a high level of phytase activity. With the increase of soluble phosphorus content in the medium, phytase activity gradually decreased. Phosphate-solubilizing fungi *Penicillium guanacastense* JP-NJ2 secretes phytase, which plays an important role in phytate mineralization [25]. White rot fungus, *Ceriporia lacerata* HG2011, secretes phytase when sodium phytate serves as the sole source of P in the culture broth, indicating that the enzyme is induced when the culture medium has low phosphatase activity [31]. Thus, extracellular phytase produced by *T. purpurogenus* can be the primary mechanism in organic phosphate mobilization.

### 4.3. Medium Carbon and Nitrogen Sources, Temperature, and pH Modulate the Phosphate Solubilization Efficiency

To further excavate the insoluble organic phosphate-mineralizing potential of *T. purpurogenus*, a categorical experimental design was used to determine the best combination of carbon and nitrogen sources, temperature, and pH to optimize P solubilization. Relwani et al. have shown that glucose and sucrose were the best C source for *Aspergillus tubingensis* for insoluble phosphate solubilization [32]. Scervino et al. reported that when glucose and $(NH_4)_2SO_4$ were used as C and N sources, insoluble phosphate solubilization was more efficient at high pH [33]. Stefanoni Rubio et al. demonstrated that glucose as the C source played a large role in influencing *T. flavus'* ability to soluble insoluble phosphate, reduce pH, and produce gluconic acid [34]. RSM has the advantages of high accuracy, low-test frequency, and short-test cycle [35]. Single-factor experiments and Central Composite Design experiments demonstrated that glucose as carbon source, $(NH_4)_2SO_4$ as nitrogen source at 28 °C, and pH of 7.0 were identified as the most effective conditions for the high organic phosphate-mineralizing activity of the *T. purpurogenus* SW-10 strain. These findings were consistent with the previous studies on the optimal parameters in enhancing the phosphate-solubilizing ability.

### 4.4. SW-10 Promotes the Growth of Different Genotypes of Maize Seedlings by Organic Phosphate Mineralization

Previous studies have reported that PSF could promote the growth of numerous plants, such as wheat, *Zea mays*, soybeans, and so on [29,36,37]. In this study, the effects of SW-10 on the growth of different genotypes of maize seedlings were different. Inbred lines of maize plants, CCM454 and 31778, were verified to be low-P tolerant and low-P sensitive, respectively [20].The dry weight of shoot and root, plant height, and P-concentration

remarkably increased when maize inbred line 31778 was inoculated with *T. purpurogenus* SW-10 strain but not in CCM454 (except P concentration). Mycorrhizal responsiveness in the modern maize genotype XY335 is higher than that of the old genotype HMY under high-P soil conditions [38]. There is increasing evidence indicating that the plant responsiveness to mycorrhizal is highly variable among different genotypes [39,40]. However, current research mostly focuses on mycorrhizal fungi. Thus, the mechanism of PSF in promoting growth in different maize genotypes demands further in-depth investigation.

## 5. Conclusions

In this study, PSF strain *T. purpurogenus* SW-10 isolated from the maize rhizosphere soil showed efficient and stable insoluble organic phosphate-mineralizing ability. The phytase produced by this strain played an important role in mineralizing organic phosphate. Single-factor experiments showed that *T. purpurogenus* had a wide range of adaptability to carbon sources, nitrogen sources, temperature, and pH. Meanwhile, *T. purpurogenus* improved plant growth and the bioavailability of phosphorus for maize plant uptake, indicating that it can be used as an effective ecofriendly P biofertilizer in sustaining agriculture and crop productivity. However, the plant responsiveness to fungi was different between maize genotypes. Further research is needed to understand the specific mechanism involved in organic phosphate mineralizing at genetic level.

**Author Contributions:** Conceptualization, X.S., F.L. and Q.S.; methodology, W.J.; software, P.Z.; validation, X.L. and Y.S.; formal analysis, X.S. and F.L.; investigation, X.S., F.L., P.Z. and Z.Z.; data curation, X.S. and F.L.; writing—original draft preparation, X.S., F.L. and Q.S.; writing—review and editing, Q.S. and W.J.; visualization, X.S. and F.L.; supervision, Y.S.; project administration, X.L.; funding acquisition, W.J. and Q.S. All authors have read and agreed to the published version of the manuscript.

**Funding:** This research was funded by the Youth Program of the Natural Science Foundation of Shandong, grant number ZR2020QC108, Science & Technology Specific Projects in Agricultural High-tech Industrial Demonstration Area of the Yellow River Delta, grant number 2022SZX24, National Natural Science Foundation of China, grant number 32201905, and Qingdao Agricultural University High-level Talents Research Foundation.

**Institutional Review Board Statement:** Not applicable.

**Informed Consent Statement:** Not applicable.

**Data Availability Statement:** Not applicable.

**Conflicts of Interest:** The authors declare no conflict of interest.

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
