# Peer review of "Talaromyces purpurogenus Isolated from Rhizosphere Soil of Maize Has Efficient Organic Phosphate-Mineralizing and Plant Growth-Promoting Abilities"

_sustainability, doi:10.3390/su15075961_

Round 1

Reviewer 1 Report

The manuscript is quite interesting and informative, and it can be accepted after minor revision. 1) The conclusion in the Abstract is not accurate, as there was no plant growth-promoting effect for the low-P tolerance inhbred acoording to your results. Please clarify it. 2) Please state the source of the low-P sensitive (31778) and low-P tolerance (CCM454) genotypes of maize, and clarify how to distinguish between low-P sensitive and low-P tolerance, what are the criteria? 3) Please clarify the method of Statistical Analysis. 4) The English should be further improved, including grammar, sentence structure, and language. 

Author Response

Dear Reviewer:

Thanks for the quick handling of our manuscript and we are pleased with the received comments. The comments are all valuable and very helpful for improving our paper. Revised portion are marked in red in the paper. The main corrections in the paper and responds to the comments are as follows:

1.The conclusion in the Abstract is not accurate, as there was no plant growth-promoting effect for the low-P tolerance inbred according to your results. Please clarify it.

Reply: Thanks for your suggestion. As suggested, we have modified the conclusion in the revised manuscript.

2. Please state the source of the low-P sensitive (31778) and low-P tolerance (CCM454) genotypes of maize, and clarify how to distinguish between low-P sensitive and low-P tolerance, what are the criteria?

Reply: Thanks for your question. The criteria for distinguishing between low-P sensitive and low-P tolerance have been described in the previous article published by our laboratory. We have added it in the Section 2.7 of the revised manuscript.

Du Q, Wang K, Xu C, et al. Strand-specific RNA-Seq transcriptome analysis of genotypes with and without low-phosphorus tolerance provides novel insights into phosphorus-use efficiency in maize. BMC Plant Biol. 2016 Oct 10;16(1):222.

3. Please clarify the method of Statistical Analysis.

Reply: Thanks for your suggestion. As suggested, we have redescribed the method of Statistical Analysis in our revised manuscript.

4. The English should be further improved, including grammar, sentence structure, and language.

Reply: Thanks for your suggestion. As suggested, we have further improved the grammar, sentence structure, and language to express more accurately in the revised manuscript.

Thank again for your comments and suggestions.

Best wishes!

Reviewer 2 Report

Rhizospheric microorganisms are known to play a significant role in solubilizing soil phosphate, which is often found in an insoluble form that is unavailable to plants. These microorganisms, which include bacteria and fungi, release organic acids and enzymes that break down the phosphate compounds in the soil, making them available for plant uptake. Overall, the use of rhizospheric microorganisms to solubilize soil phosphate has promising implications for improving plant growth and soil fertility.

This topic of this article is quite interesting for Sustainable journal. Before publishing this article, some issues have to be modified:

In Introduction section

- RMS definition is not included (line 74), but in the discussion section is in line 349.

Material and method section

- Lines 96-101

ITS pcr conditions are missing (please, add a cite)

The description of the molecular identification done in the study is not included

- Line 107 Describe molybdenum blue method and add cite

- Line 108 Add cite for Phytase activity determination. What are the units of figure 2b?

- Line 124-134. The information about RMS determination is scarce. Please, add more details of the process and also, the software used for that

- Why did the authors use  -4,14 values in Table1?

- Are data from Table 2 experimental?

Author Response

Dear Reviewer:

Thanks very much for your comments concerning our manuscript. The comments are valuable and very helpful for improving our paper. We have thought the comments carefully over and have made correction which we hope meet with your approval. Revised portion are marked in red in the paper. The main corrections in the paper and responds to the comments are as follows:

In Introduction section

1- RSM definition is not included (line 74), but in the discussion section is in line 349.

Reply: Thanks for your suggestion. As suggested, we have added the RSM definition in line 74 in the revised manuscript.

Material and method section

2- Lines 96-101 ITS PCR conditions are missing (please, add a cite). The description of the molecular identification done in the study is not included

Reply: Thanks for your attention. As suggested, we have added the PCR conditions and the description of the molecular identification.

3- Line 107 Describe molybdenum blue method and add cite

Reply: As suggested, we have added more details for molybdenum blue method and added cite in the revised manuscript.

4- Line 108 Add cite for Phytase activity determination. What are the units of figure 2B?

Reply: Sorry for this kind of mistake. As suggested, we have added cite for phytase activity determination. The units of figure 2B are U/mL. We have modified it in our revised manuscript.

5- Line 124-134. The information about RSM determination is scarce. Please, add more details of the process and also, the software used for that

Reply: As suggested, we have added more details about RSM determination and the software in the revised manuscript.

6- Why did the authors use -1.414 values in Table1? Are data from Table 2 experimental?

Reply: Thanks for your attention. The coded values of the factors are used for convenience while -1.414 are coded according to Central Composite Design (CCD).  The software automatically defines 5 levels.

Thank you again.

Best wishes!
